# Certifying Strategyproof Auction Networks

**Michael J. Curry**\*
curry@cs.umd.edu
Computer Science Department
University of Maryland
College Park, MD 20742

**Ping-Yeh Chiang**\*
pchiang@cs.umd.edu
Computer Science Department
University of Maryland
College Park, MD 20742

**Tom Goldstein**
tomg@cs.umd.edu
Computer Science Department
University of Maryland
College Park, MD 20742

**John P. Dickerson**
john@cs.umd.edu
Computer Science Department
University of Maryland
College Park, MD 20742

## Abstract

Optimal auctions maximize a seller's expected revenue subject to individual rationality and strategyproofness for the buyers. Myerson's seminal work in 1981 settled the case of auctioning a single item; however, subsequent decades of work have yielded little progress moving beyond a single item, leaving the design of revenue-maximizing auctions as a central open problem in the field of mechanism design. A recent thread of work in "differentiable economics" has used tools from modern deep learning to instead learn good mechanisms. We focus on the RegretNet architecture, which can represent auctions with arbitrary numbers of items and participants; it is trained to be empirically strategyproof, but the property is never exactly verified leaving potential loopholes for market participants to exploit. We propose ways to explicitly verify strategyproofness under a particular valuation profile using techniques from the neural network verification literature. Doing so requires making several modifications to the RegretNet architecture in order to represent it exactly in an integer program. We train our network and produce certificates in several settings, including settings for which the optimal strategyproof mechanism is not known.

## 1 Introduction

Auctions are an important mechanism for allocating scarce goods, and drive billions of dollars of revenue annually in online advertising Edelman et al. [2007], sourcing Sandholm [2007], spectrum auctions Cramton [1997], Leyton-Brown et al. [2017], and myriad other verticals Milgrom [2017], Roth [2018]. In the typical auction setting, agents who wish to bid on one or more items are presumed to have private valuations of the items, which are drawn at random from some prior valuation distribution. They then bid in the auction, possibly lying strategically about their valuation while attempting to anticipate the strategic behavior of others. The result is a potentially complicated Bayes-Nash equilibrium; even predicting these equilibria can be difficult, let alone designing auctions that have good equilibria.

This motivates the design of **strategyproof** auctions: these are auctions where players are incentivized to simply and truthfully reveal their private valuations to the auctioneer. Subject to the strategyproofness constraint, which makes player behavior predictable, it is then possible to optimize

---

the mechanism to maximize revenue. A classic strategyproof auction design is the second-price auction—coinciding with the celebrated Vickrey-Clarke-Groves (VCG) mechanism Vickrey [1961], Clarke [1971], Groves [1973]—in which participants bid only once and the highest bidder wins, but the price paid by the winner is only the amount of the second-highest bid.

In a groundbreaking work, Myerson [1981] characterized the revenue-maximizing strategyproof auction for selling one item to many bidders. However, there has been very little progress in characterizing optimal strategyproof auctions in more general settings. Optimal mechanisms are known for some cases of auctioning two items to a single bidder Manelli and Vincent [2006], Pavlov [2011], Haghpanah and Hartline [2014], Daskalakis et al. [2015]. The optimal strategyproof auction even for 2 agents buying 2 items is still unknown.

A more recent set of approaches involves formulating the auction design problem as a learning problem. Duetting et al. [2019] provide the general end-to-end approach which we build on in this paper. In brief, they design a neural network architecture to encode an auction mechanism, and train it on samples from the valuation distribution to maximize revenue. Their goal is to enforce, at least approximately, dominant-strategy incentive compatibility: a stronger notion than the Bayesian incentive compatibility of some other mechanism design approaches Cai et al. [2012a,b, 2013]. While they describe a number of network architectures which work in restricted settings, we focus on their RegretNet architecture, which can be used in settings with any number of agents or items.

The training procedure for RegretNet involves performing gradient ascent on the network inputs, to find a good strategic bid for each player; the network is then trained to minimize the difference in utility between strategic and truthful bids—this quantity is the eponymous "regret". The process is remarkably similar to adversarial training Madry et al. [2018], which applies robust optimization schemes to neural networks, and the desired property of strategyproofness can be thought of as a kind of adversarial robustness. Motivated by this connection, we use techniques from the adversarial robustness literature to compute **certifiable** upper bounds on the amount by which a player with a specific valuation profile can improve their utility by strategically lying.

While the adversarial training approach seems effective in *approximating* strategyproof auction mechanisms, neural network training is fraught with local minima and suboptimal stationary points. One can discover strategic behaviors by using simple gradient methods on RegretNet auctions, but we note that it is known from the adversarial examples literature that such results are often sub-optimal Athalye et al. [2018] and depend strongly on the optimizer Wang et al. [2019]. For this reason, it is unclear how strategyproof the results of RegretNet training are, and how much utility can be gained through strategic behavior in such auctions.

Our goal here is to learn auction mechanisms that are not only approximately strategyproof, but that come with rigorous bounds on how much they can be exploited by strategic agents, regardless of the strategy used. We achieve this by leveraging recent ideas developed for certifying adversarial robustness of neural classifiers Bunel et al. [2018], Lu and Kumar [2020], Tjeng et al. [2019], and adapting them to work within an auction framework.

**Our contributions.**

- We initiate the study of certifiably strategyproof learned auction mechanisms. We see this as a step toward achieving the best of both worlds in auction design—maintaining *provable properties* while expanding to more *general settings* than could be analyzed by traditional methods.

- We develop a method for formulating an integer-programming-based certifier for general learned auctions with additive valuations. This requires changes to the RegretNet architecture. We replace its softmax activation with the piecewise linear sparsemax Martins and Astudillo [2016], and we present two complementary techniques for dealing with the requirement of individual rationality: either formulating a nonconvex integer program, or removing this constraint from the network architecture and adding it as a learned penalty instead.

- We provide the first *certifiable* learned auctions for several settings, including a 2 agent, 2 item case where no known optimal auction exists; modulo computational scalability, our techniques for learning auctions and producing certificates for a given valuation profile work for settings with any number of items or (additive) bidders.

## 2 Background

Below, we introduce the general problem of automated mechanism design. We then describe the RegretNet approach for learning auction mechanisms, as well as the neural network verification techniques that we adapt to the auction setting. The RegretNet architecture originated the idea of parameterizing mechanisms as neural networks and training them using techniques from modern deep learning. This approach has been termed "differentiable economics", and several other papers have expanded on this approach in various settings beyond revenue-maximizing sealed-bid auctions Golowich et al. [2018], Feng et al. [2018], Manisha et al. [2018], Shen et al. [2017], Tacchetti et al. [2019].

### 2.1 Previous work

Automated mechanism design is the problem of finding good mechanisms for specific valuation distributions. In this area, one strand of work involves discretizing the space of types and solving a linear program to find the best auction in a family of mechanisms Conitzer and Sandholm [2003], Sandholm [2003]. For Bayesian incentive compatible revenue-maximizing auctions with additive bidders, Cai et al. [2012a,b, 2013] give techniques for finding the optimal mechanism, although Bayesian incentive compatibility is a weaker requirement than dominant-strategy incentive compatibility. Other work requires only access to samples from the valuation distribution over which revenue must be maximized Likhodedov and Sandholm [2004], Sandholm and Likhodedov [2015]. In this way, auction design becomes a learning problem, to which the tools of learning theory can be applied Balcan et al. [2016]. RegretNet falls into this latter family of techniques. In particular, it is an approach that learns from samples to approximate a DSIC mechanism.

### 2.2 RegretNet

In the standard auction setting, it is assumed that there are $n$ agents (indexed by $i$) buying $k$ items (indexed by $j$), and that the agents value the items according to values drawn from some distribution $P(\boldsymbol{v}_i)$. This distribution is assumed to be public, common knowledge (it is essentially the data-generating distribution). However, the specific sampled valuations are assumed to be private to each agent.

The auctioneer solicits a bid $b_{ij}$ from all agents on all items. The auction mechanism $f(\boldsymbol{b}_1, \cdots, \boldsymbol{b}_n)$ aggregates bids and outputs the results of the auction. This consists of an allocation matrix $a_{ij}$, representing each player's probability of winning each item, and a payment vector $p_i$, representing the amount players are charged. Players receive some utility $u_i$ based on the results; in this work, we focus on the case of additive utility, where $u_i = \sum_j a_{ij} v_{ij} - p_i$.

As previously mentioned, players are allowed to choose bids strategically to maximize their own utility, but it is often desirable to disincentivize this and enforce strategyproofness. The auctioneer also wants to maximize the amount of revenue paid. Duetting et al. [2019] present the RegretNet approach: the idea is to represent the mechanism $f$ as a neural network, with architectural constraints to ensure that it represents a valid auction, and a training process to encourage strategyproofness while maximizing revenue.

(We note that Duetting et al. [2019] presents other architectures, RochetNet and MyersonNet, which are completely strategyproof by construction, but only work in specific settings: selling to one agent, or selling only one item. In our work, we focus only on certifying the general RegretNet architecture.)

#### 2.2.1 Network architecture

The RegretNet architecture is essentially an ordinary feedforward network that accepts vectors of bids as input and has two output heads: one is the matrix of allocations and one is the vector of payments for each agent.

The network architecture is designed to make sure that the allocation and payments output are feasible. First, it must ensure that no item is overallocated: this amounts to ensuring that each column of the allocation matrix is a valid categorical distribution, which can be enforced using a softmax activation.

Second, it must ensure that no bidder (assuming they bid their true valuations) is charged more than the expected value of their allocation. It accomplishes this by using a sigmoid activation on the

payment output head to make values lie between 0 and 1 – call these $\tilde{p}_i$. Then the final payment for each player is $\left(\sum_j v_{ij} a_{ij}\right) \tilde{p}_i$; this guarantees that utility can at worst be 0.

Both of these architectural features pose problems for certification, which we describe below.

### 2.2.2 Training procedure

The goal of the auctioneer is to design a mechanism that maximizes the expected sum of payments received $\mathbb{E}_{\boldsymbol{v} \sim P(\boldsymbol{v})} \left[\sum_i p_i(\boldsymbol{v})\right]$, while ensuring that each player has low regret, defined as the difference in utility between the truthful bid and their best strategic bid:

$$\text{rgt}_i(\boldsymbol{v}) = \max_{\boldsymbol{b}_i} u_i(\boldsymbol{b}_i, \boldsymbol{v}_{-i}) - u_i(\boldsymbol{v}_i, \boldsymbol{v}_{-i}) \tag{1}$$

Note that this definition of regret allows only player $i$ to change their bid. However, if $\mathbb{E}_{\boldsymbol{v}}[\text{rgt}_i(\boldsymbol{v})]$ is low for all players then the mechanism must be approximately strategyproof; this is because every possible strategic bid by players other than $i$ could also be observed as a truthful bid from the support of $P(\boldsymbol{v})$.

Duetting et al. [2019] approximates regret using an approach very similar to adversarial training Madry et al. [2018]. They define a quantity $\widehat{\text{rgt}}_i$ by approximately solving the maximization problem using gradient ascent on the input – essentially finding an adversarial input for each player. Given this approximate quantity, they can then define an augmented Lagrangian loss function to maximize revenue while forcing $\widehat{rgt}$ to be close to 0:

$$\mathcal{L}(\boldsymbol{v}, \boldsymbol{\lambda}) = -\sum_i p_i + \sum_i \lambda_i \widehat{\text{rgt}}_i(\boldsymbol{v}) + \frac{\rho}{2} \left(\sum_i \widehat{\text{rgt}}_i(\boldsymbol{v})\right)^2 \tag{2}$$

They then perform stochastic gradient descent on this loss function, occasionally increasing the Lagrange multipliers $\lambda, \rho$ and recomputing $\widehat{rgt}$ at each iteration using gradient ascent. At test time, they compute revenue under the truthful valuation and regret against a stronger attack of 1000 steps. A number of high probability generalization bounds are provided for estimating regret and revenue from pointwise values on samples. With regards to the estimation of regret, we note that their generalization bound refers to the true regret at a single point (equation 1) – a quantity which a gradient-based method can only approximate but not compute exactly.

### 2.3 Mixed integer programming for certifiable robustness

Modern neural networks with ReLU activations are piecewise linear, allowing the use of integer programming techniques to verify properties of these networks. Bunel et al. [2018] present a good overview of various techniques use to certify adversarial robustness, along with some new methods. The general approach they describe is to define variables in the integer program representing neural network activations, and constrain them to be equal to each network layer's output:

$$\begin{aligned} \hat{\boldsymbol{x}}_{i+1} &= W_i \boldsymbol{x}_i + \boldsymbol{b}_i \\ \boldsymbol{x}_{i+1} &= \max(0, \hat{\boldsymbol{x}}_{i+1}) \end{aligned} \tag{3}$$

With input constraints $x_0 \in S$ representing the set over which the adversary is allowed to search, solving the problem to maximize some quantity will compute the actual worst-case input. In most cases, this is some proxy for the classification error, and the input set is a ball around the true input; in our case, computing a certificate for player $i$ involves maximizing $u_i(\boldsymbol{b}_i, \boldsymbol{v}_{-i})$ over all $\boldsymbol{b}_i \in \text{Supp}(P(\boldsymbol{v}_i))$, i.e. explicitly solving (1).

The program is linear except for the ReLU term, but this can be represented by adding some auxiliary integer variables. In particular, Tjeng et al. [2019] present the following set of constraints (supposing a $d$-dimensional layer output), which they show are feasible iff $\boldsymbol{x}_i = \max(\hat{\boldsymbol{x}}_i, 0)$:

$$\boldsymbol{\delta}_i \in \{0,1\}^d, \quad \boldsymbol{x}_i \geq 0, \quad \boldsymbol{x}_i \leq \boldsymbol{u}_i \boldsymbol{\delta}_i$$
$$\boldsymbol{x}_i \geq \hat{\boldsymbol{x}}_i, \quad \boldsymbol{x}_i \leq \hat{\boldsymbol{x}}_i - \boldsymbol{l}_i(1 - \boldsymbol{\delta}_i) \tag{4}$$

The $\boldsymbol{u}_i, \boldsymbol{l}_i$ are upper and lower bounds on each layer output that are known a priori – these can be derived, for instance, by solving some linear relaxation of the program representing the network. In particular, an approach called Planet due to Ehlers [2017] involves resolving the relaxation to compute tighter bounds for each layer in turn. Bunel et al. [2018] provide a Gurobi-based Gurobi Optimization, LLC [2020] integer program formulation that uses the Planet relaxations, later updated for use by Lu and Kumar [2020]; we modify that version of the code for our own approach. These methods output a solution which will at least be a lower bound on the true regret. Under the assumption that the chosen integer programming solver accurately reports when it has solved programs to global optimality, this will also be an upper bound – thus we will know the true maximum regret. Using bounds from Duetting et al. [2019], by computing true expected regret at many sampled points, we can estimate the overall regret of the mechanism with high probability, and we can bound the probability of sampling a point with high regret.

## 3 Techniques

These neural network verification techniques cannot be immediately applied to the RegretNet architecture directly. We describe modifications to both the network architecture and the mathematical programs that allow for their use: a replacement for the softmax activation that can be exactly represented via a bilevel optimization approach, and two techniques for coping with the individual rationality requirement. We also use a regularizer from the literature to promote ReLU stability, which empirically makes solving the programs faster.

### 3.1 Sparsemax

The RegretNet architecture applies a softmax to the network output to produce an allocation distribution where no item is overallocated. In an integer linear program, there is no easy way to exactly represent the softmax. While a piecewise linear overapproximation might be possible, we elect instead to replace the softmax with the *sparsemax* Martins and Astudillo [2016]. Both softmax and sparsemax project vectors onto the simplex, but the sparsemax performs a Euclidean projection:

$$\text{sparsemax}(x) = \arg\min_z \frac{1}{2}\|x - z\|_2^2 \text{ s. t. } 1^T z - 1 = 0, 0 < z < 1 \tag{5}$$

(Martins and Astudillo [2016] describes a cheap exact solution to this optimization problem and its gradient which are used during training. We use a PyTorch layer provided in Korrel [2020], Korrel et al. [2019].)

In order to encode this activation function in our integer program, we can write down its KKT conditions and add them as constraints (a standard approach for bilevel optimization Colson et al. [2007]), as shown in (6).

These constraints are all linear, except for the complementary slackness constraints – however, these can be represented as SOS1 constraints in Gurobi and other solvers.

The payment head also uses a sigmoid nonlinearity; we simply replace this with a piecewise linear function similar to a sigmoid.

$$(z - x) + \mu_1 - \mu_2 + \lambda 1 = 0$$
$$z - 1 \leq 0, -z \leq 0, 1^T z - 1 = 0$$
$$\mu_1 \geq 0, \mu_2 \geq 0 \tag{6}$$
$$\mu_1(z - 1) = 0, \mu_2(-z) = 0$$

### 3.2 Enforcing individual rationality

The RegretNet architecture imposes individual rationality – the requirement that no agent should pay more than they win – by multiplying with a fractional payment head, so that each player's payment is always some fraction of the value of their allocation distribution.

When trying to maximize utility (in order to maximize regret), this poses a problem. The utility for player $i$, with input bids $\boldsymbol{b}_i$, is $u_i(\boldsymbol{b}_i) = \sum_j a_{ij} v_{ij} - p_i$. The value of the allocation is a linear

combination of variables with fixed coefficients. But $p_i = \tilde{p}_i \left( \sum_j a_{ij} \boldsymbol{b}_{ij} \right)$ – this involves products of variables, which cannot be easily represented in standard integer linear programs.

We propose two solutions: we can either formulate and solve a nonconvex integer program (with bilinear equality constraints), or remove the IR constraint from the architecture and attempt to enforce it via training instead.

**Nonconvex integer programs** The latest version of Gurobi can solve programs with bilinear optimality constraints to global optimality. By adding a dummy variable, we can chain together two such constraints to represent the final payment: $p_i = \tilde{p}_i y$, and $y_i = \sum_j a_{ij} b_{ij}$. It is desirable to enforce IR constraints at the architectural level, but as described in the experiments section, it can potentially be much slower.

**Individual rationality penalty** As opposed to constraining the model architecture to enforce individual rationality constraint, we also experiment with enforcing the constraint through an additional term in the Lagrangian (a similar approach was used in an earlier version of Duetting et al. [2019]). We can compute the extent to which individual rationality is violated:

$$\mathrm{irv}_i = \max(p_i - \sum_j a_i b_i, 0) \tag{7}$$

We then allow the network to directly output a payment, but add another penalty term to encourage individual rationality:

$$\mathcal{L}(\boldsymbol{v}, \boldsymbol{\lambda}, \boldsymbol{\mu}) = -\sum_i p_i + \sum_i \lambda_i \widehat{\mathrm{rgt}}_i(\boldsymbol{v}) + \frac{\rho}{2} \left( \sum_i \widehat{\mathrm{rgt}}_i(\boldsymbol{v}) \right)^2 + \sum_i \mu_i \, \mathrm{irv}_i^2 \tag{8}$$

With this approach, the final payment no longer involves a product between allocations, bids, and payment head, so the MIP formulation does not have any quadratic constraints.

**Distillation loss** Training becomes quite unstable after adding the individual rationality penalty; we stabilize the process using distillation Hinton et al. [2015]. Specifically, we train a teacher network using the original RegretNet architecture, and use a mean squared error loss between the student and the teacher's output to train our network. The teacher may have an approximately correct mechanism, but is difficult to certify; using distillation, we can train a similar student network with an architecture more favorable for certification. Additionally, combined with the individual rationality penalty in the Lagrangian, distilling from a teacher network which enforces IR by construction also allows us to learn a student network which is discouraged from violating IR.

We allow the payments to vary freely during distillation training, to avoid vanishing gradients, and simply clip the payments to the feasible range after the training is done. Through this method, empirically, we are able to train student networks that are comparable to the teachers in performance.

### 3.3 Regularization for fast certificates

Xiao et al. [2019] point out that a major speed limitation in integer-programming based verification comes from the need to branch on integer variables to represent the ReLU nonlinearity (see Equation 4). However, if a ReLU unit is *stable*, meaning its input is always only positive or only negative, then there is no need for integer variables, as its output is either linear or constant respectively.

We adopt the approach in Xiao et al. [2019], which at train time uses interval bound propagation Gowal et al. [2018] to compute loose upper and lower bounds on each activation, and adds a regularization term $-\tanh(1 + ul)$ to encourage them to have the same sign. At verification time, variables where upper and lower bounds (computed using the tighter Planet relaxation) are both positive or both negative do not use the costly formulation of Equation 4.

# 4 Experiments

We experiment on two auction settings: 1 agent, 2 items, with valuations uniformly distributed on $[0, 1]$ (the true optimal mechanism is derived analytically and presented by Manelli and Vincent [2006]); and 2 agents, 2 items, with valuations uniformly distributed on $[0, 1]$, which is unsolved analytically but shown to be empirically learnable in Duetting et al. [2019].

For each of these settings, we train 3 networks:

- A network with a sparsemax allocation head which enforces individual rationality using the fractional payment architecture, and uses the ReLU stability regularizer of Xiao et al. [2019]

- The same architecture, without ReLU regularization

- A network that does not enforce IR, trained via distillation on a network with the original RegretNet architecture

Additionally, to investigate how solve time scales for larger auctions, we consider settings with up to 3 agents and 3 items for the architecture without IR enforcement. All training code is implemented using the PyTorch framework Paszke et al. [2019].

## 4.1 Training procedure

We generate 600,000 valuation profiles as training set and 3,000 valuation profiles as the testing set. We use a batch size of 20,000 for training, and we train the network for a total of 1000 epochs. At train time, we generate misreports through 25 steps of gradient ascent on the truthful valuation profiles with learning rate of .02; at test time, we use 1000 steps. Architecturally, all our networks use a shared trunk followed by separate payment and allocation heads; we find the use of a shared embedding makes the network easier to certify. We generally use more layers for larger auctions, and the detailed architectures, along with hyperparameters of the augmented Lagrangian, are reported in Appendix **??**.

| Auction Setting | IR | Relu Reg. | Solve time (s) | Revenue | Empirical Regret | Certified Regret | Emp./Cert. Regret |
|---|---|---|---|---|---|---|---|
| 1x2 | Yes | No | 25.6 (72.0) | 0.593 (0.404) | 0.014 (0.012) | 0.019 (0.016) | 0.731 |
| 1x2 | Yes | Yes | 7.2 (17.5) | 0.569 (0.390) | 0.003 (0.002) | 0.004 (0.003) | 0.700 |
| 1x2 | No | Yes | 0.034 (0.007) | 0.568 (0.398) | 0.009 (0.005) | 0.011 (0.004) | 0.839 |
| 2x2 | Yes | No | 13.9 (37.0) | 0.876 (0.286) | 0.009 (0.013) | 0.014 (0.016) | 0.637 |
| 2x2 (2nd) | Yes | No | 17.4 (51.9) | — | 0.007 (0.011) | 0.011 (0.013) | 0.676 |
| 2x2 | Yes | Yes | 5.8 (16.3) | 0.874 (0.285) | 0.008 (0.012) | 0.013 (0.015) | 0.626 |
| 2x2 (2nd) | Yes | Yes | 7.520 (24.2) | — | 0.008 (0.012) | 0.012 (0.014) | 0.680 |
| 2x2 | No | Yes | 5.480 (5.577) | 0.882 (0.334) | 0.006 (0.007) | 0.011 (0.011) | 0.533 |
| 2x2 (2nd) | No | Yes | 2.495 (2.271) | — | 0.011 (0.010) | 0.017 (0.017) | 0.666 |

Table 1: Summary of experimental results. Empirical regret is computed on 3000 random points and certified regret is computed on 1000 different points. (2nd) denotes the second agent in a multi-agent auction. Note that average empirical regret is only about 60-80% of the average true regret. The number in the parenthesis represents the standard deviation.

## 4.2 Results

Our results for regret, revenue and solve time are summarized in Table 1. We show the relationship between truthful and strategic bids for a learned 1 agent, 2 item mechanism in Figure 1.

**Regret certificate quality** We are able to train and certify networks with reasonably low regret – usually less than one percent of the maximum utility an agent can receive in these settings. Although mean regrets are small, the distributions are right skewed (particularly in the 2 agent, 2 item case) and there are a few points with high (approximately 0.1) regret. Crucially, we find that our certified regrets tend to be larger on average than PGD-based empirical regret, suggesting that our method reveals strategic behaviors that gradient-based methods miss.

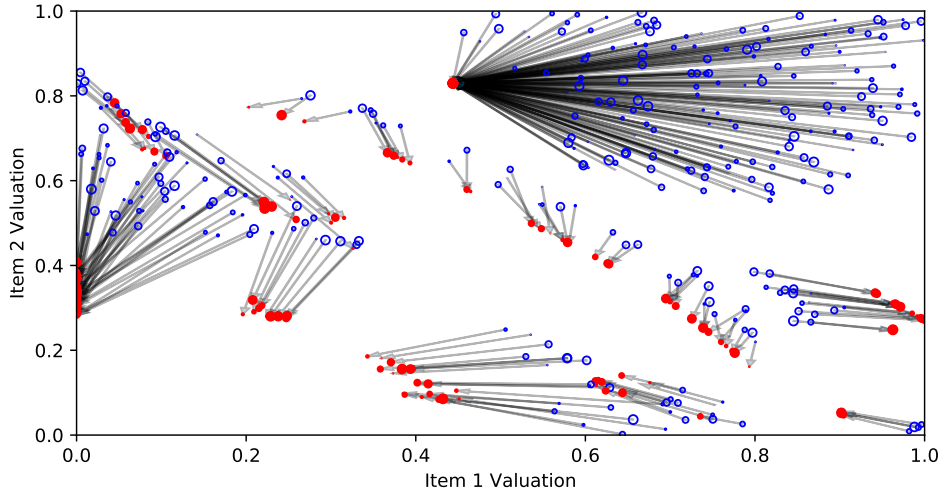

Figure 1: For the 1 agent, 2 item setting (regularized, IR enforced), this plot shows truthful bids (blue circle), with an arrow to the best strategic bid computed by the certifier (red filled). Only points with regret at least 0.005 are shown; the size of markers is proportional to the magnitude of regret. While the truthful and strategic bids are often far apart, this does not necessarily mean that violations of strategyproofness are large; in this plot, the highest regret of any point is still only 0.014.

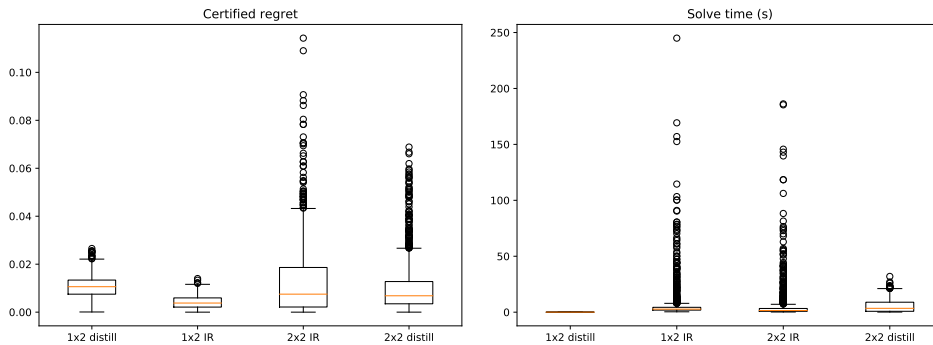

Figure 2: Certified regret and solve time for 1000 random points for IR and non-IR network architectures (regularized). Maximum utility in these settings is 2.0, so regrets are relatively small in most regions. At points with high regret, our certificates are able to detect this deficiency.

**Trained revenue**   As a baseline we consider the mean revenue from selling each item individually in a Myerson auction – in the 1 agent 2 item setting, this is 0.5; in the 2 agent 2 item setting, it is 0.8333. Our trained revenues exceed these baselines. For the 1 agent 2 item settings, the optimal mechanism Manelli and Vincent [2006] has a revenue of 0.55; our mechanisms can only exceed this because they are not perfectly strategyproof.

**Individual rationality**   Empirically, the individual rationality penalty is very effective. On average, less than 5.53% of points give an allocation violating the individual rationality constraint, and even if it is violated, the magnitude of violation is on average less than .0002 (Table **??**, Appendix **??**). Filtering out IR-violating points after the fact results in lower revenue but by less than one percent.

**Solve time**   The time required to solve the MIP is also quite important. In general, we find that ReLU stability regularization helps speed up solve time, and that solving the bilinear MIP (required for architectural enforcement of IR) is much harder than solving the mixed-integer linear program for the other architecture.

| Auction setting | Mean solve time (s) | Solve time std | Regret | Regret std |
|---|---|---|---|---|
| 2x3 | 160.66 | 142.86 | 0.0342 | 0.0169 |
| 3x2 | 5.039 | 3.40 | 0.0209 | 0.0152 |
| 3x3 | 71.81 | 54.24 | 0.0243 | 0.0204 |

Table 2: Solve times and regrets for non-IR architecture without clipped payments in larger settings on 250 random points. In general, increasing the number of items significantly slows down certification.

To investigate scalability, we also consider solve times and certified regrets for settings with larger numbers of agents and items; results are summarized in Table 2. Our experiments use the non-IR-enforcing architecture; additionally, for these experiments we do not apply hard clipping of payments. In general, increasing the number of items significantly increases the solve time – this is not too surprising, as increasing the number of items increases the dimensionality of the space that must be certified (while the same is not true for increasing the number of agents, because certificates are for one agent only). The larger solve time for 2 rather than 3 agents is harder to explain – it may simply be the result of different network properties or a more complex learned mechanism.

We note that both solve time and regret are heavily right-skewed, as shown in Figure 2. We also find that the difference between allocations, payments, and utilities computed by the actual network and those from the integer program is on the order of $10^{-6}$ – the constraints in the model correctly represent the neural network.

## 5    Conclusion and Future Work

Our MIP solution method is relatively naive. Using more advanced techniques for presolving, and specially-designed heuristics for branching, have resulted in significant improvements in speed and scalability for certifying classifiers Lu and Kumar [2020], Tjeng et al. [2019]. Our current work serves as strong proof-of-concept validation that integer-programming-based certifiability can be useful in the auction setting, and it is likely that these techniques could be applied in a straightforward way to yield a faster certifier.

The performance of our learned mechanisms is also not as strong as those presented in Duetting et al. [2019], both in terms of regret and revenue. It is unclear to us whether this is due to differences in the architecture or to hyperparameter tuning. We observe that our architecture has the capacity to represent the same class of functions as RegretNet, so we are hopeful that improved training might close the gap. The recent paper Rahme et al. [2020] finds that RegretNet is indeed very sensitive to hyperparameters, and presents an improved algorithm for auction learning which is less sensitive. The neural architectures used with this new algorithm are essentially the same as RegretNet and can also be modified to allow for certificates.

In addition to generalization bounds provided by Duetting et al. [2019], other work has dealt with the problem of estimating expected strategyproofness given only regret estimated on samples from the valuation distribution Balcan et al. [2019]. The methods presented in this work for solving the utility maximization problem have the potential to complement these bounds and techniques.

In this paper, we have described a modified version of the RegretNet architecture for which we can produce certifiable bounds on the maximum regret which a player suffers under a given valuation profile. Previously, this regret could only be estimated empirically using a gradient ascent approach which is not guaranteed to reach global optimality. We hope that these techniques can help both theorists and practitioners have greater confidence in the correctness of learned auction mechanisms.

## Broader Impact

The immediate social impact of this work will likely be limited. Learned auction mechanisms are of interest to people who care about auction theory, and may eventually be used as part of the design of auctions that will be deployed in practice, but this has not yet happened to our knowledge. We note, however, that the design of strategyproof mechanisms is often desirable from a social good standpoint. Making the right move under a non-strategyproof mechanism may be difficult for real-world participants who are not theoretical agents with unbounded computational resources. The

mechanism may impose a real burden on them: the cost of figuring out the correct move. By contrast, a strategyproof mechanism simply requires truthful reports—no burden at all.

Moreover, the knowledge and ability to behave strategically may not be evenly distributed, with the result that under non-strategyproof mechanisms, the most sophisticated participants may game the system to their own benefit. This has happened in practice: in Boston, some parents were able to game the school choice assignment system by misreporting their preferences, while others were observed not to do this; on grounds of fairness, the system was replaced with a redesigned strategyproof mechanism Abdulkadiroglu et al. [2006].

Thus, we believe that in general, the overall project of strategyproof mechanism design is likely to have a positive social impact, both in terms of making economic mechanisms easier to participate in and ensuring fair treatment of participants with different resources, and we hope we can make a small contribution to it.

## Acknowledgments

Dickerson and Curry were supported in part by NSF CAREER Award IIS-1846237, DARPA GARD, DARPA SI3-CMD #S4761, DoD WHS Award #HQ003420F0035, and a Google Faculty Research Award. Goldstein and Chiang were supported by the DARPA GARD and DARPA QED4RML programs. Additional support was provided by the National Science Foundation DMS division, and the JP Morgan Fellowship program.

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
