[Supplementary Material · supplemental.pdf]

# A Architectural and Training Details

We initialized the Lagrange multiplier of regret ($\lambda_i$) as 5, and update it every 6 batches, and we experiment with values for the constant $\rho^{\text{rgt}}$ ranging between 0.5 to 2 (reporting the choice that gave the lowest regret). For the IR violation penalty, we initialize the Lagrange multiplier of IR violation ($\mu_i$) as 20, and update the Lagrange multiplier every 6 iterations. $\mu$ is initialized as 5, and then incremented by 5 every 5 batches. For distillation, we take a mean squared error loss between the student and teacher's output, and use a multiplier of $\frac{1}{400}$. Specifically, the Lagrange multipliers are updated as follows.

$$\lambda_{i+1} = \lambda_i + \rho^{\text{rgt}}\widehat{\text{rgt}}_i \qquad\qquad \rho_{i+1}^{\text{rgt}} = \rho_i^{\text{rgt}} + \rho_{inc}^{\text{rgt}}$$
$$\mu_{i+1} = \mu_i + \rho^{\text{irv}}\,\text{irv}_i \qquad\qquad \rho_{i+1}^{\text{irv}} = \rho_i^{\text{irv}} + \rho_{inc}^{\text{irv}}$$

| Auction Setting | Inner Product | Relu Stability Regularizer | Embedding Layer |
|---|---|---|---|
| 1 Agent x 2 Items | Yes | No | 1 hidden layer x 128 units |
| 1 Agent x 2 Items | Yes | Yes | 1 hidden layer x 128 units |
| 1 Agent x 2 Items | No | Yes | 1 hidden layer x 128 units |
| 2 Agents x 2 Items | Yes | No | 2 hidden layer x 128 units |
| 2 Agents x 2 Items | Yes | Yes | 2 hidden layer x 128 units |
| 2 Agents x 2 Items | No | Yes | 2 hidden layer x 128 units |

# B Additional Experimental Information

**Hardware** All certification experiments were conducted on an AMD Ryzen 3600X CPU with 32GB RAM. Training of the network was conducted with a 2080 GPU on a university compute cluster.

**Additional experiments** Table 4 shows more detailed results for the non-IR-enforcing architecture. IR violations are relatively small, and filtering out these cases (sacrificing revenue) does not harm overall revenue too much.

Table 3 shows the results of scaling experiments for settings with more agents and items, in a setting where payment clipping is applied. Again, increasing the dimensionality of the input space by increasing the number of items seems to impose a greater cost than increasing the number of agents.

| Auction setting | Mean solve time (s) | Regret |
|---|---|---|
| 2x3 | 109.749 (159.212) | 0.027 (0.016) |
| 3x2 | 3.033 (2.377) | 0.019 (0.016) |
| 3x3 | 59.173 (53.431) | 0.022 (0.020) |

Table 3: Solve times and regrets for non-IR architecture with clipped payments in larger settings on 250 random points. In general, increasing the number of items significantly slows down certification. Standard deviations are in parentheses.

| Auction Setting | % of IR violation | Max IR violation | Mean IR violation | Revenue before enforcing IR | Revenue after enforcing IR |
|---|---|---|---|---|---|
| 1x2 | 5.53% | 0.0053 | 0.0001 (0.0003) | 0.5738 | 0.5681 |
| 2x2 | 4.60% | 0.0083 | 0.0002 (0.0007) | 0.8874 | 0.8824 |

Table 4: IR violation for the 1x2/2x2 auction settings. Note that the mean IR violation is small, and revenue after enforcing IR drops only slightly. The number in parenthesis represents the standard deviation.