[Reviews · NeurIPS 2020]

Review 1

Summary and Contributions: This paper revisits the problem of designing revenue-optimal mechanisms via deep learning. The mechanism specification is trained with the goal of maximizing revenue, for a given allocation problem and valuations drawn from fixed distributions, subject to individual rationality and incentive compatibility conditions (encoded as penalties). This paper builds on prior work by certifying the incentive compatibility using adversarial nets, which essentially strengthens the incentive compatibility guarantees and certifies those guarantees. Much of the technical work involves adapting prior network architectures for mechanism design (RegretNet) to be compatible with known verification/certification techniques. The authors evaluate the resulting trained networks empirically, and argue that their method does improve incentive properties (relative to prior methods) at the expense of revenue.

Strengths: This paper makes a solid contribution to the growing literature on mechanism design via deep learning. A downside of existing solutions is that they tend to find mechanisms that outperform known revenue bounds, presumably by relaxing incentive compatibility constraints. Since it is difficult to understand how agents will behave when faced with "almost incentive compatibility," it's important to provide tools to make this more robust. The authors make a significant step forward in that regard. The paper is written well and I enjoyed reading it. The conceptual contribution is to connect two established methods (RegretNet and adversarial verification), but the authors do a good job of describing the challenges and modifications needed to make this work and the technical contribution is non-trivial.

Weaknesses: It's left ambiguous how much of the revenue impact is due to the improved IC enforcement, and how much is due to the modified network archetecture. I would have liked to see slightly more experimentation here; e.g., testing the impact of different modifications made separately. But this is not a deal-breaker, and overall I found the empirical evaluation satisfactory.

Correctness: As far as I understand the claims and method are correct.

Clarity: The paper is well-written.

Relation to Prior Work: The relationship with prior work is clear.

Reproducibility: Yes

Additional Feedback: Post-rebuttal: I sympathize with the authors' point about it being difficult to do a fair comparison with RegretNet due to tuning issues. My overall opinion of the paper is still positive.


Review 2

Summary and Contributions: This work builds on prior work in deep learning for auctions by modifying the neural network formulation in order to admit IP based techniques for NN verification to the literature on deep learning for auctions.

Strengths: Providing guarantees for learned auctions is a key element of being able to actually use the auctions generated through machine learning (or continue to keep looking). While some of the prior models build in full strategy-proofness from the start, performance and scalability considerations may constrain methods used to those that are approximately strategyproof. This is a very promising direction and the work will likely prove to be very valuable.

Weaknesses: The work does not adequately prove that its certificates are correct. [edit - addressed in response] The work leaves open a number of questions on the scalability and cost of certification that would make it a much stronger work. Can we use this approach to adjust training times or architecture to admit stronger guarantees on empirical regret? Being able to show this would be a significant improvement. What is the performance cost of a certificate of regret? This is mentioned as an open question (as in, an open question why performance is not as good as original), but should be described within this work. Without this or the above, it is hard to extrapolate from the results included to understand broader ramifications.

Correctness: I have not verified the correctness of the approach. Additionally, the lack of a proof that the certificates are correct (even if it is a simple consequence from the literature described in section 2.2) makes it hard to verify without referring to the code.

Clarity: Much of the paper is presented as a circuit of the challenges associated with the work rather than a presentation of the results and approach. That circuit is discussed nicely and connects clearly to the literature but will be much stronger with a more direct framing of the results, and theorems surrounding correctness. Figure 1 is a very helpful representation of the scale and amounts of strategic deviation. There are informal elements that occasionally make the work hard to follow - see Line 183.

Relation to Prior Work: The paper very clearly discusses what techniques are used and from where they are derived.

Reproducibility: Yes

Additional Feedback:


Review 3

Summary and Contributions: This paper describes the training of a neural network to compute an approximately strategyproof and revenue maximizing auction with one or two buyers with additive valuations uniformly drawn from [0,1], and the case of three items and three buyers for which the IR constraint is not enforced. I have two main concerns: 1. The difference from [8] who initiated this line of inquiry is not clearly explained and in particular the experimental results obtained here are not compared with the results of [8]. It is briefly mentioned that the approach in the current paper provides certifiability but this is only very briefly discussed in section 2.2 and my impression (I am not an expert) is that this is not a novel approach but an adaptation taken from known literature. In addition section 5 mentions explicitly that the results here are “not as strong as those in [8]”. 2. There is a series of papers by Cai, Daskalakis, and Weinberg that describe how to construct linear programs that exactly solve the problems study here, especially for additive valuations. The paper does not discuss how it compares to that literature. In particular, it seems to me that it is quite straight-forward to solve the simple cases considered in this paper using the linear program approach of Cai et al. Note that that approach gives an exact optimal solution on all fronts.

Strengths: See above

Weaknesses: See above

Correctness: See above

Clarity: See above

Relation to Prior Work: See above

Reproducibility: Yes

Additional Feedback:


Review 4

Summary and Contributions: The RegretNet architecture [8] provides a way learning neural network representations of (approximately) revenue-optimal auctions. However, bounds on this degree of approximation are based on a gradient ascent optimization that does not guarantee optimality. This paper proposed a variant of the RegretNet architecture which is allows this optimization problem to be formulated as MIP and uses it to provide certified bounds on the approximation, which show that the original bounds were moderately loose but within a factor of two.

Strengths: This paper is part of a growing literature on "differentiable economics" which I am very enthusiastic about. It combines this with techniques from another active area, certifying the performance of neural networks, and I think the resulting combination will be of interest to people from both communities. The results strengthen our confidence in the accuracy of techniques like RegretNet, while also serving as a proof of concept for this type of approach that seems likely to lead to further fruitful exploration.

Weaknesses: Conceptually, I’m not entirely clear what being able to certify the exploitability at particular points buys us. The claim we can make is more rigorous, but it is still ultimately empirical. It would be nice to see some further analysis of how to translate the certificates into precise claims about the performance of the resulting mechanism. As discussed under additional feedback, the experimental evaluation, while adequate, is limited in several respects. A richer evaluation would strengthen the paper.

Correctness: All the results appear correct.

Clarity: The exposition, both conceptual and technical, is in general quite clear. There are a few points that could use further explanation or clarification, which are discussed as part of my additional feedback.

Relation to Prior Work: The coverage of prior work is good.

Reproducibility: Yes

Additional Feedback: 49, 59, 74 – The discussion here talks a bit loosely about certifying the strategyproofness of auctions. However, this is not quite what gets certified, and this is an important caveat that should be made more explicit. In particular, what gets certified is the amount by which agents can manipulate on particular type profiles. This makes no guarantees about the incentives on any other type profile, so it is a bit misleading to describe the approach as certifying the extent of strategyproofness full stop. The abstract (11) is more careful and explicit about this. 193 – I’m confused why the IR penalty approach is discussed here. Based on the description of the 3 trained networks (225-230), none of them seem to use it. However, perhaps there is something I am missing because appendix A discusses the tuning of the IR Lagrange multiplier. 201 – Perhaps related to the previous, I found the explanation of the way distillation is used a bit confusing, and didn’t really understand it until it was used in the experiments on line 229. The key piece here which is currently left implicit is that (if I have understood correctly) the teacher RegretNet architecture enforces IR by construction, while the student network does not attempt to enforce IR at all but instead simply relies on being close to the teacher to approximately enforce it. 222 – It would be nice to see some non-uniform examples, particularly anywhere this approach has the ability to shed light on still-open theory questions. 259 – What does it mean to filter out IR-violating points? 265 – Why cut off your scalability analysis at such low values? Even the largest case seems like it only took a couple of days of compute, and there certainly seems to be more room to scale the number of agents with two items. Post response: Thank you for the response. I encourage you to incorporate the discussion from it into the paper.


Review 5

Summary and Contributions: In this paper, the authors complement the recent approach of Duetting et al (ICML 2019) to compute revenue maximizing auctions for multi item settings where no theoretical approaches are known. In the initial paper, the authors are computing a regret (that they add in the Lagrangian objective function) corresponding to how the auction is incentive compatible / strategyproof This regret is computed through gradient descent There might exist a risk that the gradient descent is blocked into a local minima, such that the final auction is far from being incentive compatible (in terms of utility that strategic bidder can get by being strategic). The goal of this paper is to propose a way to compute the real value of the regret (by replacing the gradient descent by a Mixed Integer program) to increase the confidence in the correctness of the learned auction mechanism.

Strengths: Being able to certify the mechanisms learned through these new big architectures is an important problem. This paper proposes a solid compelling story to address this problem. The structure is clear and the paper well written. This paper initiates an interesting line of research on top the work of Duetting et al to compute optimal bounds on the lack of incentive compatibility. It’s also interesting to escape from a full gradient descent approach to see how other optimization pipelines can work on such new economics problem.

Weaknesses: To be able to write the optimization loss as a Mixed integer program, you have to replace the softmax (initially used by Duetting et al) by some Sparsemax. This change of architecture looks detrimental in terms of revenue since you do not recover the revenue of the initial paper of Duetting et al (that has been reproduced in some other papers). This is a problem since the ultimate goal would be to be able to obtain the optimal revenue with a certificate on the revenue. Do you have any ideas explaining this gap ?

Correctness: The empirical methodology looks fine and the authors provide their code in the submission.

Clarity: The structure is clear and the paper is well written.

Relation to Prior Work: Prior work is adequately covered.

Reproducibility: Yes

Additional Feedback:

[Author Response · NeurIPS 2020]

We thank the reviewers (**R1**, **R2**, **R3**, **R4**, and **R5**) for their thoughtful reviews, and respond to as much as we can given time and space constraints below.

**Global vs. pointwise strategyproofness** We agree the distinction between pointwise strategyproofness and global strategyproofness is an important one and thank **R4** for pointing out cases where we could further emphasize this. There is some connection between the pointwise regrets we compute and global properties of the mechanism. Duetting et al. have some generalization results that are relevant (see latest arXiv version of their paper). Their Theorem 2.2 gives a generalization bound from pointwise regret estimated on finite samples (what we compute) to true expected regret. Our networks that enforce IR satisfy the assumptions of this theorem. Given true expected regret, Lemma 2.1 allows one to bound the probability of sampling a point where regret is high – not quite a DSIC guarantee but closely related. A crucial point is that Theorem 2.2 is stated in terms of the exact pointwise regret – the true maximum of the difference in utility. It is precisely this quantity which vanilla RegretNet can only approximate but which we can compute. Making explicit reference to these results would definitely be valuable.

**Differences in performance from original RegretNet** **R1**, **R2**, and **R5** asked about the difference in empirical performance between original RegretNet and our models. Because the main goal of our work was to produce a proof of concept for certifiability, not to get SOTA performance, we made some changes from the original RegretNet architecture and training hyperparameters. Due to RegretNet's sensitivity to hyperparameters, we believe that reproducing optimal results would require a very costly hyperparameter search (for more support of this, see discussion of Rahme et al. under "Additional discussion"). Changes to enable certification include a single trunk architecture rather than separate allocation and payment networks, along with ReLU activations and sparsemax. Additionally, when training, we used different learning rates and much larger batch sizes (and therefore relatively fewer misreport updates) to make training faster. These changes might explain the performance differences. One additional point we want to emphasize is that our modified networks are not necessarily enforcing IC any more strongly than the original RegretNet – they just make it possible to detect with confidence when violations do occur after training is complete.

**Previous work in automated mechanism design** **R3** points out that more discussion of previous work in learning auctions and automated mechanism design is important. We agree with this and will add such discussion. Our underlying networks are trained using essentially the same RegretNet approach as Duetting et al; our contribution is to use this technique, but modify the network architectures to allow for exact computation of pointwise regret after training is complete. As such, much of the comparison in Duetting et al. to previous work applies to our technique as well. Specifically with regards to the Cai, Daskalakis, Weinberg papers, these are mentioned very briefly in Duetting et al and aim for Bayesian incentive compatibility (BIC), a weaker notion than dominant-strategy incentive compatibility (DSIC). RegretNet, by contrast, aims for an approximate notion of DSIC; this is what we aim for as well, while determining the presence of DSIC violations with greater confidence. We will add discussion briefly in §1 and as a new subsection in §2.

**Correctness of certificates** **R2** mentions that we do not provide or reference proofs of the correctness of our certificates. The mixed-integer formulation we use gives an exact (up to numerical error) representation of the neural network in the integer program; our certificates just consist of solving this program to find the regret-maximizing misreport. We will explicitly point to the places in the literature where the correctness of these formulations is shown (e.g. Tjeng et al. 2019). The points found as solutions give a lower bound on true regret which is often substantially higher than regret found by gradient ascent; these are also upper bounds certifying maximum true regret, under the assumption that our chosen MIP solver, Gurobi, does correctly solve problems to global optimality when it reports that it has. We will explicitly clarify this assumption as well.

**Individual rationality** **R4** raises some questions related to IR enforcement. We used both distillation from a teacher (which is perfectly IR by architectural construction) and a Lagrangian penalty to encourage the student network to be IR. We appreciate the feedback and will clarify this. Filtering out IR-violating points refers to testing network output for IR violation and if it occurs, simply charging and allocating nothing – thus player utility is zero, preserving IR, but the auctioneer revenue is also zero.

**Additional discussion** Subsequent to the NeurIPS submission deadline, a new paper was posted on arXiv, "Auction learning as a two-player game" by Rahme et al. This paper, which we feel is relevant and will discuss briefly if accepted, presents an improved training algorithm and gets better results than RegretNet on the same tasks. A core point of the paper is that RegretNet performance is very sensitive to hyperparameters (discussed throughout, see their Table 1 for specific results), a phenomenon we also observed. The architecture used for their auctioneer network is essentially the same as RegretNet (softmax, IR enforcement via product), so the modifications from our paper could be applied to it. Then the techniques we present could be used to certify networks trained using this improved algorithm, or further improved algorithms in the future.

[Meta-Review · NeurIPS 2020]

This paper received 5 reviews instead of only 3-4 because this paper kind of polarized the reviewers, so I wanted to add extra opinions to reduce the noise in the final decision. The biggest concern was that it focused only on the 2 agents / 2 item case... and we would hope to have a more complete study at the end - even if, we all agree, the problem is way more challenging. Anyway, this is a paper "in the right direction" so we happily suggest acceptance. To be clear, had it be n agents / m cases, the final outcome would certainly even be better for this paper (maybe next year ? but by 2021, the contribution might be more incremental than in 2020).